# FoxP3 Expression in Tumor-Infiltrating Lymphocytes as Potential Predictor of Response to Immune Checkpoint Inhibitors in Patients with Advanced Melanoma and Non-Small Cell Lung Cancer

**DOI:** 10.3390/cancers15061901

**Published:** 2023-03-22

**Authors:** Peter Grell, Simona Borilova, Pavel Fabian, Iveta Selingerova, David Novak, Petr Muller, Igor Kiss, Rostislav Vyzula

**Affiliations:** 1Department of Comprehensive Cancer Care, Masaryk Memorial Cancer Institute, Zluty kopec 7, 656 53 Brno, Czech Republic; 2Department of Comprehensive Cancer Care, Faculty of Medicine, Masaryk University, Kamenice 753/5, 625 00 Brno, Czech Republic; 3Department of Pathology, Masaryk Memorial Cancer Institute, Zluty kopec 7, 656 53 Brno, Czech Republic; 4Research Center for Applied Molecular Oncology (RECAMO), Masaryk Memorial Cancer Institute, Zluty kopec 7, 656 53 Brno, Czech Republic

**Keywords:** immune checkpoint inhibitors, anti-tumor immunity, predictive biomarker, malignant melanoma, NSCLC

## Abstract

**Simple Summary:**

In the last decade, immunotherapy has revolutionized the treatment of malignant melanoma and non-small cell lung cancer. Even though both tumor types have the highest tumor mutational burden, a non-negligible portion of patients do not benefit from checkpoint inhibitor treatment. The antitumor immune response is a complex multistep process of interactions between the microenvironment and the tumor. Therefore, it is crucial to seek new biomarkers to help predict the response to immunotherapy treatment. The aim of our study was to describe tumor tissue, its microenvironment, and immune profile, and correlate it with the response to treatment.

**Abstract:**

Immune checkpoint inhibitors (ICI) are the main therapy currently used in advanced malignant melanoma (MM) and non-small cell lung cancer (NSCLC). Despite the wide variety of uses, the possibility of predicting ICI efficacy in these tumor types is scarce. The aim of our study was to find new predictive biomarkers for ICI treatment. We analyzed, by immunohistochemistry, various cell subsets, including CD3+, CD8+, CD68+, CD20+, and FoxP3+ cells, and molecules such as LAG-3, IDO1, and TGFβ. Comprehensive genomic profiles were analyzed. We evaluated 46 patients with advanced MM (31) and NSCLC (15) treated with ICI monotherapy. When analyzing the malignant melanoma group, shorter median progression-free survival (PFS) was found in tumors positive for nuclear FoxP3 in tumor-infiltrating lymphocytes (TILs) (*p* = 0.048, HR 3.04) and for CD68 expression (*p* = 0.034, HR 3.2). Longer PFS was achieved in patients with tumors with PD-L1 TPS ≥ 1 (*p* = 0.005, HR 0.26). In the NSCLC group, only FoxP3 positivity was associated with shorter PFS and OS. We found that FoxP3 negativity was linked with a better response to ICI in both histological groups.

## 1. Introduction

Over the last decade, treatment with ICIs has transformed the prognosis of patients with locally advanced or metastatic MM or NSCLC. It has brought numerous treatment options, even in the first line, from monotherapies to combinations with other ICIs, such as CTLA-4 (cytotoxic T lymphocyte antigen 4) or LAG-3 (Lymphocyte-activation gene 3) inhibitors, or with chemotherapy in NSCLC. However, without strong biomarkers, the decision-making process is becoming more and more difficult. CheckMate 067, with a 6.5-year follow-up, proved remarkable overall survival (OS) in patients with advanced malignant melanoma treated with nivolumab+ipilimumab (NIVO+IPI) or nivolumab monotherapy. Nevertheless, still, more than 40% of patients treated with the NIVO+IPI combination did not reach significant tumor regression [1]. In the NSCLC, the greatest benefit from monotherapy with a PD-1/PD-L1 inhibitor was found in patients with high PD-L1 expression (≥50%) [2,3,4]. Despite the high PD-L1 expression, half of the patients in this highly selected population still did not achieve a significant therapeutic response [3].

Both malignant melanoma and NSCLC are considered high mutation burden cancer types and typically show high initial ICI responses [5]. However, the effect of immunotherapy relies not only on the tumor mutational burden, but also on the complex network between tumor cells and the tumor microenvironment (TME) [6]. Thus, the TME represents a subject of increasing interest as a potential source of predictive factors for treatment with ICIs or even as an additional therapeutic target. It comprises a heterogeneous population of cancer cells, immune cells, vessels, stroma, signaling mediators, and extracellular matrix proteins [7]. The complexity of the TME in the immune response manifests the variety of immune cells and markers, both pro- and anti-inflammatory functions.

The importance of TILs in responsiveness to ICIs was demonstrated in ALK and EGFR mutated NSCLC tumors, where very low rates of co-localized PD-L1 expression and CD8+ TILs were linked with low ORR [8]. A meta-analysis of 2559 cancer patients across the tumor types proved that high CD8+ TILs rates were associated with longer PFS in melanoma and the NSCLC subgroup. However, in certain patients, the CD8 T cells positivity does not predict the response to treatment. This may be explained by different subtypes of CD8 T cells—central, effector, stem-like, and tissue-resident memory cells [9]. Furthermore, not only functional status (dysfunction, exhaustion) but differences in spatial distribution are contributing to T cell antitumor effector function [10]. Another important immune cell group expressing CD68+ is tissue-associated macrophages (TAM), which promote phagocytosis and mediate the recruitment and activation of other immune cells. High expression levels of CD68 in tumors correlated with an adverse prognosis [11]. Another immune cell population potentially contributing to ICI therapy failure are regulatory lymphocytes (Tregs). Tregs are functioning as central mediators of immune function. Furthermore, the expression of the transcription factor Forkhead box protein P3 (FoxP3) in regulatory T cells (Tregs) may have utility as a predictor of response to ICIs. B cells express many pro- and anti-inflammatory factors essential for tumor inflammation. Studies suggest that B cell infiltration may predict response to ICIs and contribute to immune-related adverse events [12,13].

The expression of other immune inhibitory markers, such as LAG-3 [14], T-cell immunoglobulin mucin-3 (TIM-3) [15], and indoleamine 2, 3-dioxygenase 1 (IDO1) [16], has been associated with resistance to ICI therapy. The anti-LAG-3 drug relatlimab in combination with nivolumab has already proven its efficacy in patients with malignant melanoma in a phase 2–3 trial and is more effective than nivolumab monotherapy [17]. The IDO1 inhibitor epacadostat showed promising results in early clinical trials but failed to prove efficacy in a phase 3 trial in advanced melanoma (ECHO-301/Keynote-252 study) [18]. Another signaling pathway contributing to ICI resistance can be driven by transforming growth factor beta (TGFβ). Co-administration of TGFβ-blocking and anti-PD-L1 antibodies reduced TGFβ signaling in stromal cells, facilitated T-cell penetration into the center of the tumor, and activated antitumor immunity and tumor regression. This suggests that TGFβ participates in the tumor microenvironment and restricts T cell infiltration [19].

Based on previously written, we conducted a prospective study to investigate different subpopulations of immune cells, such as CD3+ cells, CD8+ cells, CD68+ cells, CD20+ cells, and FoxP3+ cells, and the expression of molecules, including LAG-3, IDO1, and TGFβ, by immunohistological staining in patients with advanced malignant melanoma and NSCLC, and assessed their role in ICI effectiveness. Furthermore, we analyzed comprehensive genomic profiles and evaluated their contribution to the efficacy of ICI treatment in patients with advanced malignant melanoma and NSCLC.

## 2. Materials and Methods

We prospectively enrolled patients with advanced or metastatic malignant melanoma, or NSCLC, treated with anti-PD-1 immune checkpoint inhibitor (nivolumab—administered intravenously at a dose of 3 mg/kg every 2 weeks or at a dose of 240 mg or pembrolizumab—administered intravenously 200 mg every 3 weeks) from 2017 to 2021 at Masaryk Memorial Cancer Institute in Brno, Czech Republic. The inclusion criteria were as follows: advanced/metastatic disease, measurable disease by RECIST criteria, planned treatment with anti-PD-1 checkpoint inhibitor monotherapy, expected survival of more than 3 months, and ability to understand and the willingness to sign a written informed consent document. Informed consent was obtained from each participating subject. The study was approved by the Institutional Ethic Committee of Masaryk Memorial Cancer Institute, reference number 2017/1890/MOU, 27 June 2017.

Immunohistochemistry (IHC) staining was performed using formalin-fixed and paraffin-embedded tissue specimens. Specifically, one block from the tumor resection with a maximally representative tumor population (containing an invasive tumor growth front) was evaluated by a pathologist highly experienced in the evaluation of tissue analysis and blinded to the patient’s characteristics or treatment outcomes. This method of assessment is very close to real clinical practice. The mismatch repair (MMR) protein status was determined by using monoclonal antibodies provided by DAKO: MLH-1 (clone ES05), MSH-2 (clone FE11), MSH-6 (clone EP49), and PMS-2 (clone EP51). Null expression in at least one of the MMR proteins with a positive control in non-tumor cells was considered a deficiency. For staining for CD3, we used antibody clone SP7 by DCS—Innovative Diagnostik-Systeme (Hamburg, Germany) for CD8 clone SP16 by Thermo Scientific (Fremont, CA, USA). We evaluated stromal and intraepithelial infiltration separately. For stromal characteristics, we assessed the percentage of tumor stroma formed by lymphocytes (the sample was considered positive if at least 10% of cells demonstrated positive staining by immunohistochemistry for CD3 or CD8). For intraepithelial evaluation, we estimated the average number of lymphocytes in direct contact with tumor cells per area of one high magnification, counted manually (the sample was considered positive if at least 10 cells were positive for CD3 or CD8 staining). For CD20 staining, we used antibody clone L26 by DAKO (Stockholm, Sweden), and membrane positivity was evaluated and scored on a qualitative scale from 0 to 3 (score 0 and 1 was considered negative, score 2 a 3 as positive). For the CD68 staining clone, KP1 by DAKO was used, and plasmatic positivity was scored from 0 to 3. For PD-L1 staining, we used antibody clone 22C3 by DACO and scores using CPS and TPS used in clinical practice. For PD1 staining, clone NAT105 by Abcam (Cambridge, UK) was used, and membrane positivity was scored from 0 to 3. LAG-3 staining was performed by antibody LAG-3 clone BLR028F by Novus Biologicals (Littleton, CO. USA), and nuclear positivity was scored from 0 to 3. For IDO1 staining, we used antibody clone 1 A3 by LifeSpan BioSciences (Seattle, WA, USA), and plasmatic positivity was scored from 0 to 3. For TGFβ, we used antibody clone EPR21143 by Abcam. For TGFβ, we evaluated immune and tumor cells separately and scored on a semiqualitative scale from 0 to 3. For FoxP3 antibody clone SP97 (by Abcam) was used, nuclear positivity was evaluated, and estimated the number of lymphocytes with nuclear positivity per area of one high power field (objective 40×, ocular 10×/22 mm) of view and scored as positive if more than 10 lymphocytes were stained positive. Examples of different sections are shown in Figure 1.

Comprehensive genomic profiling was performed on the patient with available tissue of sufficient quality. Tumor tissue dissected from FFPE blocks was used for mutational analysis. DNA was isolated using the QIAamp DNA FFPE Tissue Kit (Qiagen, Manchester, UK). The sequencing library was prepared using the TruSight Oncology 500 DNA kit according to the recommended protocol. Sequencing was performed on an Illumina NextSeq500 instrument using pair-end sequencing (2 × 101 bp). Mutations, MSI status and TMB, were determined using the tumor-only workflow of TSO500 (Illumina Inc., San Diego, CA, USA). For the TMB status cut-off, 10 muts/Mb was used.

Factors evaluated in association with progression-free survival and overall survival included age at the onset of checkpoint inhibitors treatment, type of cancer, line of therapy, baseline laboratory values (white blood cell count, absolute neutrophil count, absolute monocyte count, absolute lymphocyte count, neutrophil to lymphocyte ratio (NLR), C-reactive protein (CRP) level, lactate dehydrogenase (LDH) level, drug use (metformin, proton pump inhibitors), treatment toxicity, immunohistochemistry expression of different markers (MMR status, CD3 stromal expression, CD3 intraepithelial expression, CD8 stromal expression, CD8 intraepithelial expression, CD20, CD68, PDL-1, PD1, LAG3, IDO1, TGFβ in immune cell expression, TGFβ in tumor cells, and FoxP3 nuclear expression), and TMB. Response to therapy was evaluated by using RECIST criteria version 1.1. PFS was defined as the time from the beginning of checkpoint inhibitors therapy to the first documented objective disease progression or death. OS was defined as the time from the beginning of checkpoint inhibitors therapy to death due to any cause. Clinical benefit from treatment was defined as achieving complete or partial response or at least six months of stable disease. Frequency analysis and summary statistics were used to characterize the sample data set. Survival curves were estimated using the Kaplan–Meier method. A log-rank test was used to test the difference between survival curves (PFS or OS) for different factors. All point estimates include 95% confidence intervals (CIs). Fisher’s exact or Chi-squared tests were used to establish the significance of the association between categorical variables. The Cox proportional hazard model was used to calculate hazard ratios (HR). All statistical analyses were performed employing R 4.2.0 and a significance level of 0.05.

## 3. Results

### 3.1. Patient Characteristics

Forty-six patients were enrolled and evaluated for our analysis. Patients’ characteristic is summarized in Table 1. The median age was 68 years, and 11 patients were female (24%). Histological types of tumors were as follows: 31 patients with malignant melanoma and 15 patients with non-small lung cancer. Most of the patients started treatment with ICI as a first line of treatment (36 patients, 78%), a second line in eight patients (17%), and a third or later line two patients (4.4%). Patients were treated with nivolumab (all patients with malignant melanoma, 10 patients with NSCLC) or pembrolizumab (five patients with NSCLC).

The median follow-up was 37 months. Disease progression was identified in 31 patients (67.4%), and 18 patients (39.1%) were still alive. There was no statistically significant difference in PFS according to tumor type. Median PFS for malignant melanoma was 11.0 months (95% CI 4.8, NR; not reached) and for lung carcinoma 8.2 months (95% CI 5.1–NR). We found no statistically significant difference in OS according to tumor type. The median OS was 27.0 months (95% CI 20.0–NR) for malignant melanoma, 15.0 months (95% CI 9.3–NR) for lung carcinoma. The best response to treatment was complete response in 10 patients (23.0%), partial response in 14 (32.0%), stable disease in two patients (4.5%), and disease progression in 18 (41.0%). In two patients, the treatment effect was not evaluated due to treatment toxicity. Twenty-five patients achieved clinical benefit from treatment (56.8%).

Comprehensive genomic profiling was performed on 31 patients (26 with malignant melanoma and five with NSCLC). All assessed gene variants are summarized in Table 2. High TMB was found in 19 patients (61.2%), and the median muts/Mb was 15.3. No patient had a tumor with microsatellite instability. The pathogenic *BRAF* variant was found in 14 patients (45.2%, in malignant melanoma patients only), *NRAS* in 12 patients (38.7%), *TP53* in 8 (25.8%), *KRAS* in 3 (9.7%), *ARID2* in 3 (9.7%), and *CDKN2 A* in 3 (9.7%); other pathogenic variants were less common.

### 3.2. Correlation with Survival Parameters in the Malignant Melanoma Group

In patients with malignant melanoma, we found shorter PFS related to FoxP3 nuclear positivity (17.0 vs. 4.5 months, *p* = 0.048, HR 3.04, 95% CI 0.95–9.71) (Figure 2A) and CD68 positivity (15.0 vs. 4.1 months, *p* = 0.034, HR 3.21, 95% CI 1.02–10.1). Longer PFS was achieved in patients with tumors with PD-L1 TPS ≥ 1 (NR vs. 5.5 months, *p* = 0.005, HR 0.26, 95% CI 0.06–0.69). A trend toward longer PFS was found in patients with high TMB (*p* = 0.058) and PD-L1 CPS 1 and more (*p* = 0.052). No factors were associated with longer OS in malignant melanoma; the trend for longer OS was found in tumors with PD-L1 TPS 1 and more (*p* = 0.06). Results for malignant melanoma are summarized in Table 3.

### 3.3. Correlation with Survival Parameters in the NSCLC Group

In patients with NSCLC, shorter PFS was found in tumors with FoxP3 nuclear positivity (14.8 vs. 1.8 months, *p* = 0.003, HR 8.7, 95% CI 1.55–48.7) (Figure 2B). A trend to longer PFS was found in tumors with CD8 IEL negativity (*p* = 0.089). Worse OS was associated with FoxP3 nuclear positivity (22.0 vs. 8.3 months, *p* = 0.035, HR 3.86, 95% CI 1.01–14.8), and a trend toward longer OS was found in tumors with TGFβ IC negativity (*p* = 0.066) and a high TMB (*p* = 0.063). Results for NSCLC are summarized in Table 4.

### 3.4. Co-Expression of FoxP3 and CD68

Based on our results that positive FoxP3 and CD68 expression are significantly related to shorter progression-free survival, we also examined the predictive value of the co-expression of FoxP3 and CD68. There was a correlation between FoxP3 and CD68 expression in the melanoma group (*p* = 0.004) (Table 5), but no correlation in the NSCLC group (*p* = 0.999) (Table 6). We analyzed patients with positivity at least in one parameter (FoxP3 or CD68; 10 patients) and with negative expression in both markers (34 patients). We found significantly longer PFS in patients with tumors with negativity in both parameters in patients with malignant melanoma (median PFS 17.0 vs. 3.9 months, *p* = 0.008, HR 3.97, 95% CI 1.33–11.9; Figure 3A) and NSCLC (median PFS 19.0 vs. 2.3 months, *p* < 0.001, HR 18.8, 95% CI 2.11–167; Figure 3B). Furthermore, we found a significantly shorter median OS in the tumors with positivity in at least one parameter in the NSCLC group (*p* = 0.020), but not in malignant melanoma group (results are summarized in Table 7). 

### 3.5. Correlation of IHC Expression with Clinical Parameters

We found no correlation between clinical parameters (age, sex, type of tumor, metastatic sites, line of treatment, using metformin, proton pump inhibitors) and IHC expression of different markers. In addition, we found no correlation between baseline laboratory parameters (LDH, levels of leukocytes, neutrophils, monocytes, and lymphocytes) and IHC expression, except CRP (C-reactive protein) levels. Positive expression of FoxP3 was significantly associated with elevated (>5 mg/L) CRP levels (*p* = 0.024). The correlation between FoxP3 expression and baseline CRP levels is summarized in Table 8. IHC expressions of different markers were not associated with the type of tumor except for the expression of TGFβ IC, which was predominantly negative in lung carcinoma (*p* = 0.048).

## 4. Discussion

We found that patients with nuclear FoxP3 positive expression in TILs achieved shorter PFS when treated with PD-1 inhibitors. The predictive value was confirmed independently for malignant melanoma and NSCLC. Although FoxP3 is considered the master transcription factor for Tregs, it can also lead to reprogramming conventional T cells into Tregs [20]. The high expression of FoxP3 is found in a specific subtype of Tregs, the effector Tregs, which are activated and highly suppressive [21]. Tregs suppress excessive immune responses to maintain immune homeostasis. In the TME, Tregs are involved in tumor development and progression by inhibiting antitumor immunity through inhibition of costimulatory signals, interleukin (IL)-2 consumption, secretion of inhibitory cytokines, metabolic modulation of tryptophan and adenosine, and direct killing of effector T cells [22]. Furthermore, in the TME, FoxP3 is also expressed in tumor cells. In the majority of tumor types, the localization of FoxP3 expression is predominantly nuclear, besides the cytoplasmatic expression in pancreatic cancer [23,24,25,26].

The prognostic value of FoxP3 expression is very controversial, depending on the site of expression (tumor cells or TILs) and the histopathological tumor type. Regarding the TILs, increased infiltration of FoxP3+ Tregs was associated with improved OS in colorectal, head and neck, and esophageal cancer, whereas in melanoma, lung, cervical, renal, hepatocellular, gastric, and breast cancers, it was linked with shorter OS [27]. On the other hand, while the higher count of FoxP3-positive tumor cells predicts better survival in gastric and prostate cancers [24,28,29], in melanoma and NSCLC, high expression was associated with an unfavorable clinical prognosis, leading to shorter overall survival and recurrence-free survival [30,31]. However, none of these studies analyzed the prognostic value of FoxP3 specifically in patients treated with ICI. In contrast to these results from previously mentioned studies, our study has confirmed FoxP3 expression as a prognostic factor only in the NSCLC group, but not in melanoma patients.

Regarding immunotherapy with ICIs, the impact of PD-1 inhibitors on Tregs is not yet fully clear. In mouse models, the selective PD-1 deficiency in Tregs increased their suppressive function. Moreover, the lack of PD-1 signaling suppressed effector T cell activation, expansion, and cytokine production more effectively than wild-type Tregs [32]. In clinical studies, high infiltration of Tregs was found in patients treated with PD-1 inhibitors who achieved disease hyper-progression [33]. New FoxP3 inhibitors augmented antitumor immunity and provided a therapeutic benefit in cancer models [34]. Another checkpoint, the CTLA-4, is highly expressed not only in activated T cells but also in Tregs. CTLA-4 loss or inhibition in mice models resulted in reduced Treg function contributing to antitumor responses by anti-CTLA-4 treatment [35]. Moreover, in malignant melanoma, the clinical benefit of ipilimumab correlated with the decreased number of intratumoral Tregs [36].

In the present study, regardless of tumor subtype, patients with an abundance of FoxP3-positive lymphocytes in TME experienced worse benefit from monotherapy with PD-1 inhibitors, suggesting that FoxP3 is a predictive factor for treatment with ICIs. These results, corresponding with studies on animal models, support the rationale for combination therapy in malignant melanoma and lung cancer with high expression of FoxP3. As CTLA-4 inhibitors reduce the number and function of Tregs in malignant melanoma, the positivity of FoxP3 could help in the decision-making process between monotherapy with PD-1 inhibitors and combination therapy with PD-1 and CTLA-4 inhibitors. However, confirmation with a larger number of patients is needed.

The second marker whose positivity was significantly linked with shorter PFS in patients with malignant melanoma was CD68 expression. CD68 is overexpressed in tumor-associated macrophages (TAM). Macrophages’ roles in cancer are complex. Activated macrophages are often classified as pro-inflammatory M1 macrophages or anti-inflammatory M2 macrophages. M2-like macrophages support angiogenesis and tumor growth, contributing to tumor aggressiveness. In the TME, these cells can recruit Tregs and inhibit effector T cells by secreting IL-10 and expressing PD-L1 [37,38]. This recruitment of Tregs by TAMs can explain the significant correlation between FoxP3 and CD68 expression in our study. TAMs’ mechanisms of primary resistance to PD-1 inhibitors have been demonstrated in preclinical studies. After administration, the anti-PD-1 antibody binds to tumor-infiltrating T cells at an early stage but is subsequently captured by TAMs due to the presence of Fcγ receptors, ultimately leading to drug failure [39]. Confirming these findings in a clinical setting, we identified high CD68 expression in malignant melanoma as a negative predictor for monotherapy with PD-1 inhibitors.

Moreover, by combining two negative biomarkers, FoxP3 (Tregs) and CD68 (TAMs), we can select an even more specific group of patients who benefited from treatment with PD-1 inhibitors the most, and in the melanoma subgroup, we confirmed its predictive value but not prognostic. In the NSCLC subgroup, CD68 overexpression did not prove to be predictive nor prognostic value. This can be explained by the small number of patients in the subgroup. Nonetheless, the negative expression in both biomarkers proved its positive predictive and prognostic value also in NSCLC.

Regarding the laboratory findings and IHC expressions, patients with FoxP3-positive tumors have significantly higher CRP levels. Indeed, all patients with FoxP3 positivity have elevated CRP levels. This correlation has been studied in other tumor types. In colorectal cancer, an inverse relationship was found between the systemic inflammatory response (elevated CRP with cut-off 10 mg/L) and FoxP3+ Tregs infiltration in the intratumor stroma [40]. On the other hand, renal cell carcinoma patients with strong infiltration of Foxp3+ lymphocytes had significantly higher CRP levels (elevated CRP with cut-off 5 mg/L). The biological explanation for these contradictory findings between melanoma, renal, lung carcinoma, and colorectal cancer remains unclear, but we can see a similar pattern between increased infiltration of FoxP3+ Tregs and OS, as mentioned above.

Additionally, we studied currently used predictors, such as PD-L1 expression and TMB. PD-L1 TPS positivity (≥1) was associated with better PFS in patients with malignant melanoma but not in patients with NSCLC. PD-L1 predictive efficacy depends on the assay used, different thresholds, and tumor type specificity. Indeed, the results from previous studies are very contradictory. In one study with malignant melanoma, PD-L1 tumor cell positivity proved its predictive value in response to ICIs [41]. In the pivotal phase 3 trial in metastatic melanoma, PD-L1 positivity on tumor cells was associated with better treatment response, though responses were also seen in PD-L1 negative tumors [42]. Nevertheless, in the large meta-analysis, high PD-L1 expression did not correlate with OS or PFS [43].

We are aware of the certain limitations of our study. Although our study was prospective and patients were uniformly treated with PD-1 monotherapy, it was limited to a relatively low number of enrolled patients. Our results were driven by two diagnoses—malignant melanoma and NSCLC. Moreover, regarding the heterogeneity of our study group, we must interpret our results consciously. Nevertheless, melanoma and lung cancer surely share common characteristics proven in previous studies, such as high average TMB and FoxP3 expression in tumor cells or TILs representing negative prognostic factors.

Furthermore, a significant proportion of patients in the NSCLC group were pretreated, which may have influenced the identification and validation of different predictive biomarkers taking into account the dynamics of changes in cancer immunity interpersonally and intrapersonally. A limited number of markers could limit or simplify the view into a complex antitumor microenvironment. However, the methodology of our study was designed to represent real daily clinical practice, allowing an easier implementation into a daily clinical routine and easier confirmation of our results.

## 5. Conclusions

In summary, we demonstrated that high FoxP3 expression in TILs has a negative predictive value for therapy with PD-1 inhibitors in malignant melanoma and NSCLC. The expression of CD68+ lymphocytes proved its predictive value only in malignant melanoma. Nonetheless, combining these two biomarkers could predict the response to PD-1 inhibitors with higher probability in both tumor groups. We confirmed that other biomarkers, such as high PD-L1 expression, although not universally but in certain diseases, could predict the response.

## Figures and Tables

**Figure 1 cancers-15-01901-f001:**
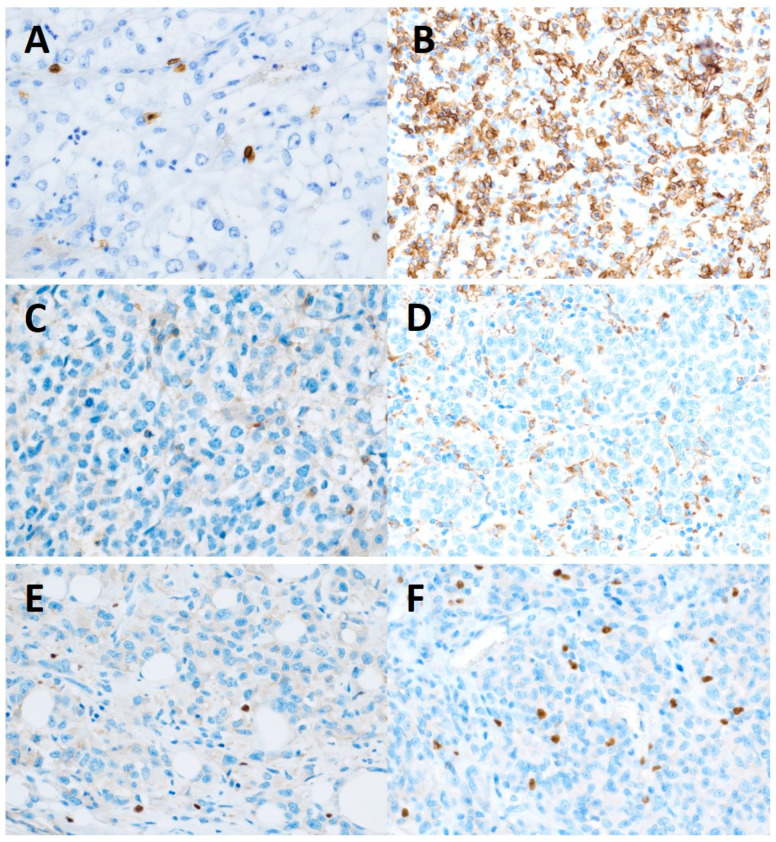
Immunohistochemical staining of samples under microscope (magnification: 200×). (**A**) CD8 intraepithelial expression low. (**B**) CD8 intraepithelial expression high. (**C**) Negative expression of CD68. (**D**) Positive expression of CD68. (**E**) Negative expression of FoxP3. (**F**) Positive expression of FoxP3.

**Figure 2 cancers-15-01901-f002:**
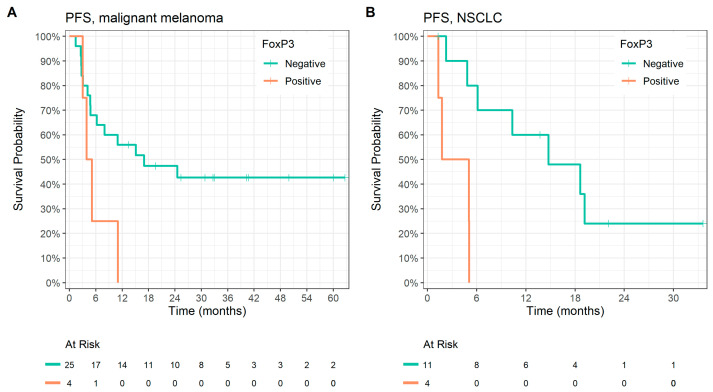
Kaplan–Meier analysis of progression-free survival according to FoxP3 expression in (**A**) malignant melanoma, median PFS was 17.0 vs. 4.5 months, *p* = 0.048; (**B**) NSCLC, median PFS was 14.8 vs. 1.8 months, *p* = 0.003.

**Figure 3 cancers-15-01901-f003:**
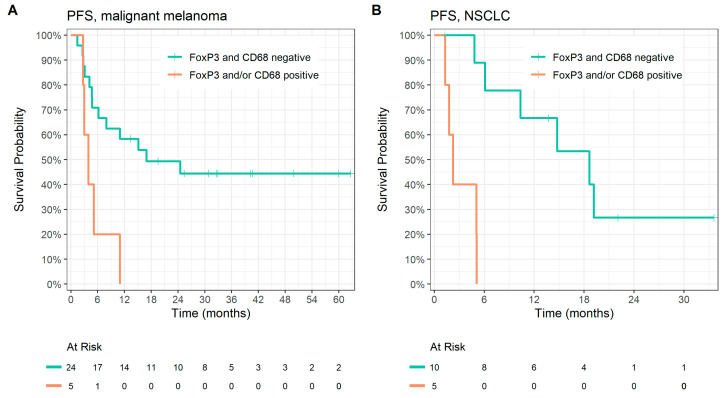
Kaplan–Meier analysis of progression-free survival according to FoxP3 and CD68 co-expression in (**A**) malignant melanoma, median PFS 17.0 vs. 3.9 months, *p* = 0.008; and (**B**) NSCLC, median PFS 19.0 vs. 2.3 months, *p* < 0.001.

**Table 1 cancers-15-01901-t001:** Patient characteristics.

Variable	Overall N = 46	Malignant Melanoma N = 31	NSCLC N = 15
Age (years)			
Median (IQR)	68 (62, 73)	70 (64, 76)	65 (59, 71)
Range	43, 85	52, 85	43, 79
Sex			
Women	11 (24%)	8 (26%)	3 (20%)
Men	35 (76%)	23 (74%)	12 (80%)
Lymphocytes (×10^9^/L)			
≤0.8	6 (13%)	4 (13%)	2 (13%)
>0.8	40 (87%)	27 (87%)	13 (87%)
Leukocytes (×10^9^/L)			
≤10	37 (80%)	28 (90%)	9 (60%)
>10	9 (20%)	3 (9.7%)	6 (40%)
Neutrophils (×10^9^/L)			
≤7	39 (85%)	29 (94%)	10 (67%)
>7	7 (15%)	2 (6.5%)	5 (33%)
Monocytes (×10^9^/L)			
≤1.2	40 (87%)	26 (84%)	14 (93%)
>1.2	6 (13%)	5 (16%)	1 (6.7%)
LDH (μkat/L)			
≤3.55 (Men), 3.75 (Women)	31 (78%)	18 (72%)	13 (87%)
>3.55 (Men), 3.75 (Women)	9 (22%)	7 (28%)	2 (13%)
Missing	6	6	0
CRP (mg/L)			
≤5	18 (49%)	14 (64%)	4 (27%)
>5	19 (51%)	8 (36%)	11 (73%)
Missing	9	9	0
Metformin comedication			
Yes	4 (8.7%)	1 (3.2%)	3 (20%)
No	42 (91.3%)	30 (96.8%)	12 (80%)
PPI comedication			
Yes	7 (15.2%)	2 (6.4%)	5 (33.3%)
No	39 (84.8%)	29 (93.6%)	10 (66.6%)
Line of treatment			
1st line	36 (78%)	30 (97%)	6 (40%)
2nd line	8 (17%)	0 (0%)	8 (53%)
3rd or later line	2 (4.4%)	1 (3.2%)	1 (6.7%)
Best overall response			
Complete response	10 (23%)	9 (29%)	1 (7.7%)
Partial response	14 (32%)	8 (26%)	6 (46%)
Stable disease	2 (4.5%)	1 (3.2%)	1 (7.7%)
Disease progression	18 (41%)	13 (42%)	5 (38%)
Unknown	2	0	2
Clinical benefit rate	25 (56.8%)	18 (58.1%)	7 (54%)
Unknown	2	0	2
Survival parameters (median, 95% CI)			
PFS (months)	10 (5.1, 19)	11 (4.8, —)	8.2 (5.1, —)
OS (months)	25 (18, —)	27 (20, —)	15 (9.3, —)

Abbreviation: NSCLC, non-small cell lung cancer; PFS, progression-free survival; OS, overall survival; LDH, lactate dehydrogenase; CRP, C-reactive protein; PPI, proton-pump inhibitors. Laboratory values normal vs. above upper limit of normal.

**Table 2 cancers-15-01901-t002:** Results of comprehensive genomic profiling and pathogenic gene variants.

Gene Variant	All Patients		Malignant Melanoma		NSCLC	
*BRAF*	14	45.2%	14	53.8%	0	0.0%
*NRAS*	12	38.7%	12	46.2%	0	0.0%
*TP53*	8	25.8%	5	19.2%	2	40.0%
*KRAS*	3	9.7%	1	3.8%	2	40.0%
*ARID2*	3	9.7%	0	0.0%	0	0.0%
*CDKN2A*	3	9.7%	3	11.5%	0	0.0%
*CTNNB1*	2	6.5%	2	7.7%	0	0.0%
*PTEN*	2	6.5%	0	0.0%	2	40.0%
*ARID1A*	1	3.2%	1	3.8%	0	0.0%
*ATM*	1	3.2%	0	0.0%	1	20.0%
*POLE*	1	3.2%	1	3.8%	0	0.0%
*SF3B1*	1	3.2%	1	3.8%	0	0.0%

Abbreviation: NSCLC, non-small cell lung cancer.

**Table 3 cancers-15-01901-t003:** Association of assessed IHC markers with survival parameters in the melanoma group.

		PFS			OS		
Characteristic	N	HR	Median Survival	*p*-Value	HR	Median Survival	*p*-Value
CD3 IEL				0.202			0.122
Negative	13	—	—		—	—	
Positive	16	1.87	8.8		2.27	25	
CD3 stromal				0.667			0.834
Negative	6	—	9.5		—	47	
Positive	22	0.78	15		1.15	26	
CD8 IEL				0.484			0.093
Negative	17	—	11		—	—	
Positive	12	1.38	10		2.31	22	
CD8 stromal				0.811			0.539
Negative	7	—	11		—	13	
Positive	22	1.14	9.5		0.70	27	
CD20				0.966			0.417
Negative	26	—	9.5		—	31	
Positive	4	0.97	13		1.68	25	
CD68				**0.034**			0.968
Negative	26	—	15		—	27	
Positive	4	3.21	4.1		1.03	36	
FoxP3				**0.048**			0.852
Negative	25	—	17		—	27	
Positive	4	3.04	4.5		1.15	33	
IDO1				0.519			0.180
Negative	23	—	8.0		—	26	
Positive	7	0.70	15		0.38	—	
LAG-3				0.606			0.770
Negative	23	—	11		—	31	
Positive	7	0.75	15		1.18	25	
TGFβ IC				0.384			0.366
Negative	16	—	5.5		—	27	
Positive	14	0.67	16		0.63	—	
TGFβ TC				0.916			0.991
Negative	26	—	11		—	27	
Positive	4	0.92	33		1.01	39	
PD1				0.850			0.501
Negative	22	—	11		—	31	
Positive	8	1.10	10		1.44	25	
PD-L1 CPS				0.052			0.794
<1	9	—	6.2		—	27	
≥1	17	0.39	17		0.87	25	
PD-L1 CPS				0.195			0.529
<10	19	—	8.0		—	26	
≥10	7	0.45	—		0.67	—	
PD-L1 CPS				0.407			0.240
<50	22	—	9.5		—	25	
≥50	4	0.54	30		0.31	—	
PD-L1 TPS				**0.005**			0.060
<1	18	—	5.5		—	20	
≥1	9	0.20	—		0.32	—	
PD-L1 TPS				0.268			0.467
<10	24	—	9.5		—	25	
≥10	3	0.34	—		0.48	—	
PD-L1 TPS				0.268			0.467
<50	24	—	9.5		—	25	
≥50	3	0.34	—		0.48	—	
TMB high				0.058			0.923
Negative	9	—	4.4		—	27	
Positive	17	0.41	15		0.95	26	

Abbreviation: IEL, intraepithelial; IC, immune cell; TC tumor cell; CPS, combined positive score; TPS, tumor proportion score; TMB, tumor mutational burden. Median survival in months.

**Table 4 cancers-15-01901-t004:** Association of assessed IHC markers with survival parameters in the NSCLC group.

		PFS			OS		
Characteristic	N	HR	Median Survival	*p*-Value	HR	Median Survival	*p*-Value
CD3 IEL				0.767			0.980
Negative	7	—	6.1		—	14	
Positive	8	1.20	10		1.02	19	
CD3 stromal				0.289			0.258
Negative	4	—	4.9		—	14	
Positive	11	0.46	13		0.47	22	
CD8 IEL				0.089			0.122
Negative	9	—	14		—	19	
Positive	5	2.86	4.8		2.60	9.3	
CD8 stromal				0.131			0.423
Negative	7	—	22		—	15	
Positive	7	2.75	5.1		1.67	11	
CD20				0.522			0.701
Negative	13	—	8.2		—	15	
Positive	2	1.66	10		0.67	17	
CD68				0.097			0.182
Negative	14	—	10		—	17	
Positive	1	5.98	2.3		4.14	8.5	
FoxP3				**0.003**			**0.035**
Negative	11	—	15		—	22	
Positive	4	8.70	3.4		3.86	8.3	
IDO1				0.767			0.601
Negative	13	—	8.2		—	15	
Positive	2	1.27	10		1.51	13	
LAG-3				0.146			0.293
Negative	11	—	13		—	19	
Positive	4	2.47	3.3		1.92	12	
TGFβ IC				0.115			0.066
Negative	13	—	13		—	19	
Positive	2	3.47	3.7		4.35	7.0	
TGFβ TC *							
Negative	15						
Positive	0						
PD1				0.807			0.693
Negative	12	—	6.1		—	14	
Positive	3	1.18	15		1.31	22	
PD-L1 CPS				0.162			0.222
<1	1	—	—		—	—	
≥1	14		6.1			14	
PD-L1 CPS				0.481			0.553
<10	5	—	19		—	14	
≥10	10	1.73	8.2		0.69	19	
PD-L1 CPS				0.439			0.437
<50	9	—	14		—	19	
≥50	6	1.60	5.5		1.63	13	
PD-L1 TPS				0.313			0.712
<1	6	—	19		—	16	
≥1	9	1.98	6.1		0.80	15	
PD-L1 TPS				0.313			0.712
<10	6	—	19		—	16	
≥10	9	1.98	6.1		0.80	15	
PD-L1 TPS				0.557			0.649
<50	10	—	10		—	16	
≥50	5	1.45	4.8		1.34	15	
TMB high				0.199			0.063
Negative	3	—	6.1		—	11	
Positive	2	0.00	21		0.00	23	

Abbreviation: IEL, intraepithelial; IC, immune cell; TC tumor cell; CPS, combined positive score; TPS, tumor proportion score; TMB, tumor mutational burden; * No patient was positive for TGFβ TC expression. Median survival in months.

**Table 5 cancers-15-01901-t005:** Correlation between FoxP3 and CD68 expression in malignant melanoma group.

	CD68 Expression			
	Negative	Positive	Total	*p*-Value ^1^
FoxP3 nuclear expression				0.004
negative	24 (96%)	1 (4.0%)	25 (100%)	
positive	1 (25%)	3 (75%)	4 (100%)	
total	25 (86%)	4 (14%)	29 (100%)	

^1^ Fisher’s exact test.

**Table 6 cancers-15-01901-t006:** Correlation between FoxP3 and CD68 expression in NSCLC group.

	CD68 Expression			
	Negative	Positive	Total	*p*-Value ^1^
FoxP3 nuclear expression				0.999
negative	10 (91%)	1 (9.1%)	11 (100%)	
positive	4 (100%)	0 (0%)	4 (100%)	
total	14 (93%)	1 (6.7%)	15 (100%)	

^1^ Fisher’s exact test.

**Table 7 cancers-15-01901-t007:** Correlation between FoxP3 and CD68 expression and survival parameters.

		PFS			OS		
Characteristic	N	HR	Median Survival	*p*-Value	HR	Median Survival	*p*-Value
FoxP3 + CD68 (melanoma group)				**0.025**			0.517
Both Negative	24	—	17		—	27	
Positive	5	3.97	3.9		1.55	15	
FoxP3 + CD68 (NSCLC group)				**0.001**			**0.020**
Both Negative	10	—	19		—	23	
Positive	5	18.8	2.3		5.24	8.5	

**Table 8 cancers-15-01901-t008:** Correlation between FoxP3 expression and baseline CRP levels.

	CRP				
	≤5 mg/L	>5 mg/L	Missing	Total	*p*-Value
FoxP3 expression					0.024
Negative	17 (47%)	13 (36%)	6 (17%)	36 (100%)	
Positive	0 (0%)	6 (75%)	2 (25%)	8 (100%)	
Missing	1 (50%)	0 (0%)	1 (50%)	2 (100%)	
Total	18 (39%)	19 (41%)	9 (20%)	46 (100%)	

Abbreviations: FoxP3; Forkhead box protein P3; CRP, C-reactive protein.

## Data Availability

Data will be made available from the corresponding author on reasonable request.

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
