# Peer review of "FoxP3 Expression in Tumor-Infiltrating Lymphocytes as Potential Predictor of Response to Immune Checkpoint Inhibitors in Patients with Advanced Melanoma and Non-Small Cell Lung Cancer"

_cancers, 2023, doi:10.3390/cancers15061901_

Round 1
Reviewer 1 Report
The authors Grell et al, studied the role of FoxP3 and CD68 and proved that they are the potential predictors of response to immune checkpoint inhibitors in patients with advanced solid tumor. They found that the absence of expression of several biomarkers, such as CD68 and FoxP3, is associated with better survival. The study presented is convincing and novel, some aspects of the manuscript need improvements. I suggest authors to address the following comments.
Major Comments
1. Weaknesses include the superficial nature of the description of the study in the introduction. Similarly, the Discussion section also needs improvement by highlighting the key findings and their implications.
Author Response
Dear reviewer,
We would like to thank you for your comments. We have improved the Introduction and Discussion section. Based on comments from other reviewers, we have made major changes to the article, the biggest change being to focus only on malignant melanoma and NSCLC. Therefore, all parts of the manuscript were changed, including the introduction and discussion.
Sincerely
Peter Grell
Reviewer 2 Report
In this prospective study, the authors used clinical patients’ tumor samples and information to do a very comprehensive analysis. They used multiple lab techniques (IHC staining, DNA sequence, tumor-only workflow) and different statistical methods to evaluate the correlation of different factors (immunological markers, TMB, lab results etc.) and PFS/OS with the treatment of ICIs. The study is of significant clinical relevance.
However, this study cannot be published without the below listed modifications:
1. The structure of the manuscript is confusing:
§ The subtitles jumped from 3.1 to 3.3 directly. And there are two 3.2.
§ Starting from line 198 to line 212, highly suggest the authors to make a Table to this data too.
§ Starting from line 213, another summarized table should be made for the genomic profiling results.
§ From line 252, ‘using metformin’, ‘proton pump inhibitors’ should be combined into Table 1, and the lab results should move forward to right after Table 1, also shown in a summarized table. The correlation between patients’ characteristics, lab results and IHC expression should be described in manuscript after IHC expression relevance. ‘Correlation of clinical parameters’ can be deleted.
§ There are 2 Table 3s.
§ Figure 2-4 are not very necessary since they just repeated the information in Tables. They can be deleted or moved to supplementary data.
2. In line 149, ‘the immunohistochemistry assessment was performed by pathologist highly experienced in the evaluation of tissue analysis and blinded to the patient's characteristics or treatment outcomes’. The assessment should be performed by at least two independent pathologists.
3. The most important part, it is too sketchy to draw the conclusion of the importance of CD68 and FoxP3. 1) More figures or tables should be shown to interpret the co-expression of CD68 and FoxP3 better. 2) More clinical patients should be recruited for further prospective study or maybe retrospective study as well. 3) If possible, some in vivo experiments to test the hypothesis will be helpful too.
Author Response
Dear reviewer,
we would like to thank you for your comments. We appreciate your suggestions and are sure they will contribute to a better manuscript. Based on comments from other reviewers, we have made significant changes to the article. The major change is to focus only on malignant melanoma and NSCLC. Therefore, all parts of the manuscript were revised.
1 The structure of the manuscript:
- The subtitles jumped from 3.1 to 3.3 directly. And there are two 3.2.
We made significant changes, including correct numbering.
- Starting from line 198 to line 212, highly suggest the authors to make a Table to this data too.
For better reading, we have put the data in a table.
- Starting from line 213, another summarized table should be made for the genomic profiling results.
We have created a table for genomic data.
- From line 252, ‘using metformin’, ‘proton pump inhibitors’ should be combined into Table 1, and the lab results should move forward to right after Table 1, also shown in a summarized table. The correlation between patients’ characteristics, lab results and IHC expression should be described in manuscript after IHC expression relevance. ‘Correlation of clinical parameters’ can be deleted.
We added data (used drugs, lab results) into the Table 1. The correlation between patients’ characteristics, lab results and IHC expression were evaluated and are now described in the manuscript. We found correlation between CRP level and FoxP3 expression.
- There are 2 Table 3s.
We corrected table numbering.
- Figure 2-4 are not very necessary since they just repeated the information in Tables. They can be deleted or moved to supplementary data.
We removed several figures and added figurese for FoxP3/CD68 co-expression.
- In line 149, ‘the immunohistochemistry assessment was performed by pathologist highly experienced in the evaluation of tissue analysis and blinded to the patient's characteristics or treatment outcomes’. The assessment should be performed by at least two independent pathologists.
It can be objected that the evaluation carried out in this way (i.e. qualified estimation) is always burdened with some individual (subjective) error and to eliminate it, it would be necessary to have it evaluated by several pathologists. The proposed procedure (two pathologists) can only remove outlier errors (individual oversight of a feature), but not a systematic error. Especially if the pathologists are from the same workplace, it is very likely that they will systematically evaluate a feature with a similar error common to that workplace. In real clinical practice when foe example HER2 or PD-L1 is evaluated, it is also evaluated by one person experienced in the given matter. It is not evaluated by two or more pathologists, and yet a potentially curative or toxic therapy or very expensive treatment is indicated based on this (equally subjective error-laden) evaluation. This method of evaluation is similar to real clinical practice.
- The most important part, it is too sketchy to draw the conclusion of the importance of CD68 and FoxP3. 1) More figures or tables should be shown to interpret the co-expression of CD68 and FoxP3 better. 2) More clinical patients should be recruited for further prospective study or maybe retrospective study as well. 3) If possible, some in vivo experiments to test the hypothesis will be helpful too.
We made significant changes to this part of manuscript, added tables and figures. We prospectively enrolled patients described in our manuscript, unfortunately a large number of patients had to be excluded due to insufficient tissue quality. But we understand that a larger patient population makes the results more robust. We are planning a new study focusing on FoxP3 and CD68 expression with a larger number of patients. An in vivo experiment with FoxP3 and CD68 and immunotherapy would be very interesting, but also very difficult to perform and unfortunately it is not within the capabilities of our institution.
Sincerely
Peter Grell
Reviewer 3 Report
The present manuscript “FoxP3 and CD68 as potential predictors of response to immune checkpoint inhibitors in patients with advanced solid tumor” provided by Grell et al. addresses an interesting and clinically relevant topic.
Even though it is interesting, interpretation of the data provided is not possible in the current form. Data presentation and statistical traceability are the major issues that have to be fixed. The manuscript conception needs thorough revision.
Major concerns:
1) The specimen collection is somehow odd. What is the idea in adding tumor entities of n=1?
The manuscript must focus on malignant melanoma. NSCLC data may be added, but with a very critical view on it.
2) Therefore all other tumors (including renal) must be omitted.
3) This also demands a focus of the text on the tumors used and no global statements on “solid tumors” per se.
4) For the interpretation of the data it is also of importance how many slides you used for each patients sample in IHC. Did you examine different areas of each tumor? How were sample specimen obtained? By biopsy or from tumors after resection? These informations must be given with the statistics of the IHC data.
5) In the tables 3-4 (by the way “Table 3” is used for two tables), for each median calculated, the number of individuals that contributed to this median (n=?) must be given.
6)What does NR mean in Table 3 (melanoma) and 4? Please state in the legend.
7) Also for the Figs. 2, 3 and 4, each survival line needs the number of individuals that contributed. Additionally, “n” should be given in the legend.
8) Conclusion must only cover presented data. Please focus on melanoma and NSCLC
Author Response
Dear reviewer,
we would like to thank you for your comments. We appreciate your suggestions and are sure they will contribute to a better manuscript. Based on your comments we have made significant changes to the article. The major change is to focus only on malignant melanoma and NSCLC. Therefore, all parts of the manuscript were revised.
1) The specimen collection is somehow odd. What is the idea in adding tumor entities of n=1?
The manuscript must focus on malignant melanoma. NSCLC data may be added, but with a very critical view on it.
2) Therefore all other tumors (including renal) must be omitted.
3) This also demands a focus of the text on the tumors used and no global statements on “solid tumors” per se.
Very important comments. We initially designed our study as tumor agnostic, i.e. include all patients treated with immunotherapy regardless of tumor type. There are several indications that immunotherapy with checkpoint inhibitors works across different histological tumor types. On the other hand, its effectiveness varies with respect to the type of tumors. We were aware that certain diagnoses would dominate the cohort and discussed this aspect of the study a lot, but at the same time we wanted to include patients with other tumor types. Unfortunately, only a small number of patients other than malignant melanoma and NSCLC were enrolled and included in the analysis. For more conclusive data interpretation, we decided to exclude other diagnoses from the manuscript and included only patients with malignant melanoma and NSCLC.
4) For the interpretation of the data it is also of importance how many slides you used for each patients sample in IHC. Did you examine different areas of each tumor? How were sample specimen obtained? By biopsy or from tumors after resection? These informations must be given with the statistics of the IHC data.
We evaluated one block from the tumor resection with a maximally representative representation of the tumor population, containing an invasive growth front (description added to manuscript). We are aware of tumor heterogeneity. It is common even within the scope of one particular slide and is at least as significant a problem as heterogeneity in different parts of the tumor. We evaluated the slides in this way, which is similar to real clinical practice.
5) In the tables 3-4 (by the way “Table 3” is used for two tables), for each median calculated, the number of individuals that contributed to this median (n=?) must be given.
We corrected the numbering of the tables, added the number of patients to the table and figures.
6)What does NR mean in Table 3 (melanoma) and 4? Please state in the legend.
We have reworked the tables, including descriptions of abbreviations.
7) Also for the Figs. 2, 3 and 4, each survival line needs the number of individuals that contributed. Additionally, “n” should be given in the legend.
We added “n¨ into the figures.
8) Conclusion must only cover presented data. Please focus on melanoma and NSCLC
We have changed the concept of the manuscript to address this issue.
Sincerely
Peter Grell
Round 2
Reviewer 2 Report
After revision, Peter Grell et al. addressed some of the previous issues. However, this study still cannot be published because of following:
1. The introduction is pretty hard to read, there are several paragraphs that are just repeating similar background information, which makes the introduction redundant and confusing. For example, the first three paragraphs talk about the difficulties and limitations of ICI treatment in MM and NSCLC, which can be combined into just one paragraph.
2. Previously the authors discussed prediction markers in general solid tumors including NSCLC and melanoma. The revised version only included NSCLC and melanoma. The context of research changed. NSCLC and melanoma are two separate malignant cancers that need to be discussed independently. The analysis of ‘whole group’ throughout the entire revised paper should be deleted.
3. There is no further discussion or evaluation of genomic profiling results, can be deleted or moved to Supplementary Data.
4. Results 3.5 can be deleted or moved to Supplementary Data.
5. TMB was being discussed and showed significance too, but TMB was not included in the conclusions. There should be some explanation for that.
6. FoxP3 represents Tregs, CD68 represents TAM. The importance of these two markers has already been discussed extensively. Without more experiments or prospective study with novelty, the revised version is still not qualified for publication in a journal like Cancers.
Author Response
Dear reviewer,
Thank you for your comments. We would like to respond to them.
- The introduction is pretty hard to read, there are several paragraphs that are just repeating similar background information, which makes the introduction redundant and confusing. For example, the first three paragraphs talk about the difficulties and limitations of ICI treatment in MM and NSCLC, which can be combined into just one paragraph.
We have made changes to introductions and we reduced the text as recommended.
- Previously the authors discussed prediction markers in general solid tumors including NSCLC and melanoma. The revised version only included NSCLC and melanoma. The context of research changed. NSCLC and melanoma are two separate malignant cancers that need to be discussed independently. The analysis of ‘whole group’ throughout the entire revised paper should be deleted.
The research context has not changed. We only removed patients or diseases with low numbers and focused on results that allow us to make sufficiently valid statements. Indeed, it is not possible to draw universal conclusions based on the outcome of the treatment of one patient with a specific diagnosis (N=1).
- There is no further discussion or evaluation of genomic profiling results, can be deleted or moved to Supplementary Data.
The result of TMB was also part of the genomic profiling and its relation to survival parameters is evaluated in the manuscript. We added a note to the text that other genomic profiling results did not relate to survival.
- Results 3.5 can be deleted or moved to Supplementary Data.
We found a correlation between CRP level and expression of FoxP3. This finding is relatively new, as we have not found any other work on malignant melanoma or NSCLC that has found this association. CRP generally is associated with worse survival in patients with malignant melanoma treated with ICIs. And FoxP3 expression may be one of the explanations. That's why we think it's important to mention.
- TMB was being discussed and showed significance too, but TMB was not included in the conclusions. There should be some explanation for that.
We included TMB in discussion and conclusion. The reason why we hesitated to put TMB in the conclusion was that we wanted to mention only the novel results of our work.
- FoxP3 represents Tregs, CD68 represents TAM. The importance of these two markers has already been discussed extensively. Without more experiments or prospective study with novelty, the revised version is still not qualified for publication in a journal like Cancers.
We agree that the importance of these two markers has already been extensively discussed. However, the overwhelming majority of work focused on the prognostic significance of these markers in various cancer types. Our work is unique and novel in that we have shown that these markers are related to the effectiveness of checkpoint inhibitors immunotherapy and are therefore the predictive biomarkers that we so fundamentally need for this treatment modality.
Although our study was prospective and clinically oriented, we agree that our results must be verified on a larger group of patients. Our study generates a hypothesis for the potential use of these biomarkers. On other hand, the landmark study by Le et al on PD-1 Blockade in Tumors with Mismatch-Repair Deficiency published in the New England of Medicine in 2015 had only 41 patients but revolutionized the whole field of oncology. Similarly, study with checkpoint inhibitors in early MSI-H rectal cancer published last year had only 12 patients (!!! and also published in NEJM) and rewrote the treatment guidelines. Therefore, we think that our study is suitable for publication in such a prestigious journal as Cancers.
Reviewer 3 Report
The revised manuscript by Grell et al. has improved significantly .
There is one point that needs to be addressed:
in line 171 the Authors deleted "nivolumab". However the drug used in the study must be stated. The study is useless without exact specification of the drugs used.
In case of patients studies, also a brief generalized description of the treatment regiment should be given. However, I'm aware that this may differ between patients...
Author Response
Dear reviewer,
thank you for your comments.
We added treatment specifications. All patients were treated with anti-PD-1 therapy. Patients were treated with nivolumab (all patients with malignant melanoma, 10 patients with NSCLC) or pembrolizumab (5 patients with NSCLC). The dosing is also described in Material and Methods.
Regards
Peter Grell
Round 3
Reviewer 2 Report
Please delete all the results discussing MM and NSCLC together as ‘whole group’, including Tables, Figures, and manuscripts.
Author Response
Dear reviewer,
Thank you for your commnets.
we made all the suggested changes and removed the "whole group" analysis.
Sincerely
Peter Grell